# Atomic Force Microscope Nanoindentation Analysis of Diffuse Astrocytic Tumor Elasticity: Relation with Tumor Histopathology

**DOI:** 10.3390/cancers13184539

**Published:** 2021-09-10

**Authors:** Abraham Tsitlakidis, Anastasia S. Tsingotjidou, Aristeidis Kritis, Angeliki Cheva, Panagiotis Selviaridis, Elias C. Aifantis, Nicolas Foroglou

**Affiliations:** 1First Department of Neurosurgery, AHEPA University Hospital, Aristotle University of Thessaloniki, 54636 Thessaloniki, Greece; panagiotis.selviaridis@gmail.com (P.S.); nforoglou@auth.gr (N.F.); 2Laboratory of Anatomy, Histology and Embryology, School of Veterinary Medicine, Aristotle University of Thessaloniki, 54124 Thessaloniki, Greece; astsing@vet.auth.gr; 3Laboratory of Physiology, School of Medicine, Aristotle University of Thessaloniki, 54124 Thessaloniki, Greece; kritis@auth.gr; 4Department of Pathology, School of Medicine, Aristotle University of Thessaloniki, 54124 Thessaloniki, Greece; antacheva@yahoo.gr; 5Laboratory of Mechanics and Materials, Polytechnic School, Aristotle University of Thessaloniki, 54124 Thessaloniki, Greece; mom@mom.gen.auth.gr

**Keywords:** diffuse glioma, elastic modulus, atomic force microscopy, WHO grade, IDH, tissue mechanics

## Abstract

**Simple Summary:**

Biomechanics has emerged as a key player in diffuse glioma progression and invasion, as the glioma cells interact both chemically and mechanically with the extracellular matrix to facilitate proliferation and cell migration, altering the mechanical properties of tumor and adjacent brain tissue. The quantification of these properties is expected to contribute to advances in glioma biology, diagnostic instrumentation, and treatment technology. Previous studies have associated isocitrate dehydrogenase gene family (IDH) mutations and World Health Organization (WHO) grade with differences in glioma elasticity, although not on fresh tissue specimens. This is the first study to investigate the combined influence of glioma IDH mutation status and WHO grade on both tumor and peritumoral white matter fresh tissue elasticity by using atomic force microscope (AFM) nanoindentation. It is also the first to systematically determine the elastic modulus of human white matter using AFM nanoindentation with spherical tips on fresh tissue slices ex vivo.

**Abstract:**

This study aims to investigate the influence of isocitrate dehydrogenase gene family (IDH) mutations, World Health Organization (WHO) grade, and mechanical preconditioning on glioma and adjacent brain elasticity through standard monotonic and repetitive atomic force microscope (AFM) nanoindentation. The elastic modulus was measured ex vivo on fresh tissue specimens acquired during craniotomy from the tumor and the peritumoral white matter of 16 diffuse glioma patients. Linear mixed-effects models examined the impact of tumor traits and preconditioning on tissue elasticity. Tissues from IDH-mutant cases were stiffer than those from IDH-wildtype ones among anaplastic astrocytoma patients (*p* = 0.0496) but of similar elasticity to IDH-wildtype cases for diffuse astrocytoma patients (*p* = 0.480). The tumor was found to be non-significantly softer than white matter in anaplastic astrocytomas (*p* = 0.070), but of similar elasticity to adjacent brain in diffuse astrocytomas (*p* = 0.492) and glioblastomas (*p* = 0.593). During repetitive indentation, both tumor (*p* = 0.002) and white matter (*p* = 0.003) showed initial stiffening followed by softening. Stiffening was fully reversed in white matter (*p* = 0.942) and partially reversed in tumor (*p* = 0.015). Tissue elasticity comprises a phenotypic characteristic closely related to glioma histopathology. Heterogeneity between patients should be further explored.

## 1. Introduction

Mechanical interactions play a central role in diffuse glioma progression and invasion, as the extracellular matrix (ECM) provides mechanical stimuli to glioma cells, serving as a substrate to adhere and facilitate migration [1,2]. Concurrently, glioma cells actively remodel normal brain ECM to their benefit and disrupt the tissue’s mechanical homeostasis [3,4]. Neurosurgeons have long attributed the resultant alteration of the tissue mechanical properties to differences in consistency between white matter and glioma tissue. However, the quantification of such differences has the potential to further illuminate gliomagenesis [5,6]; to assist in the development and refinement of diagnostic techniques that utilize knowledge about tissue mechanical behavior; and to incorporate information about tissue mechanics in aspects of glioma surgery (e.g., neuronavigation [7] and haptics for surgical robotics [8,9,10]). To acquire such information, the study of the mechanical behavior of diffuse glioma tissues under internal mechanical stress is needed both at the microscale and the macroscale.

Tissues are non-linear/non-homogeneous composite materials and their mechanical behavior can be described by using non-linear elasticity and more complex theories. However, under low deformation, they may be assumed to behave as homogeneous linear elastic materials, thus making related experimental interpretation more convenient and feasible. The mechanical stress *σ* (force per unit area) that such materials experience during compression/tension is taken to be proportional to the strain *ε* (length change per initial length) through the elastic modulus *E* (*σ* = *E**ε*), measured in Pascal (Pa) [11]. In sensitive and soft materials, such as glioma tissues, elasticity is usually probed using elastography or atomic force microscope (AFM) nanoindentation. Elastography maps tissue elasticity in vivo at the macroscale using imaging techniques through wave propagation interpretations. Both ultrasound elastography [12,13,14,15] and magnetic resonance elastography (MRE) [16,17,18,19] have been extensively used for the determination of glioma elasticity. The AFM [20], on the other hand, is a type of scanning probe microscope designed for local measurements, whereby a specimen moves under a microcantilever. During the process of the imposed continuous contact between the cantilever tip and the specimen, the cantilever bends and its deflection is recorded along with the specimen position. Consequently, mapping of the specimen surface morphology and/or measurement of the specimen elasticity is performed with nanoscale resolution for both position and force. Due to its resolution, versatility, and ability to provide direct measurements of tissue elasticity, AFM nanoindentation became an established technique for the ex vivo determination of fresh unfixed brain [21,22,23,24,25] and glioma tissue elasticity [26,27].

Two histopathological factors that determine diffuse glioma prognosis, namely WHO grade [12,13,14,16,17,18,27] and isocitrate dehydrogenase gene family (IDH) mutation status [16,27], have been studied and associated with the observed differences in diffuse glioma tissue and peritumoral white matter elasticity. However, the possibility of a synergistic effect on tissue elasticity has not yet been examined ex vivo on fresh tissue at the microscale. This study primarily aims to determine the combined influence of these two factors on glioma and peritumoral white matter elasticity by directly probing the mechanical behavior of fresh tissues ex vivo. A secondary objective pursued is the investigation of the mechanical behavior of diffuse glioma and white matter tissue under repetitive indentation.

## 2. Materials and Methods

### 2.1. Tissue Acquisition

The present work is a prospective observational cross-sectional study for investigating the influence of IDH mutation status and WHO grade on diffuse glioma tissue and peritumoral white matter elasticity, as determined ex vivo on fresh surgical tissue specimens with AFM nanoindentation. The work has been approved by the institutional bioethics committee and all procedures followed were in accordance with the Helsinki Declaration. Prior written informed consent was obtained by all patients (or a parent/guardian, when the patient could not be held accountable for consent).

All patients of age > 16 years submitted to craniotomy for the resection of a supratentorial intra-axial tumor by the same surgeon (N.F.) between 2017 and 2019 were eligible for inclusion in the study. Demographic data (patient age and gender) were recorded preoperatively. Fresh tissue specimens were acquired from the periphery of the resected lesions during the operation, avoiding necrotic regions and tissue compression. All procedures were performed according to the rule of maximal safe tumor resection. The study did not influence and was not influenced by the extent of resection or the treatment protocol. The specimens for the study comprised a minimal part of the resected tumor. They were obtained after securing sufficient tissue specimens for histopathology, which was also conducted as a separate procedure following the usual route. In shortage of sufficient resectable tissue, the patient was excluded from the study.

White matter tissue specimens, resected at 1 cm from the lesion border on FLAIR and/or T1W contrast-enhanced MRI, were kept during the surgical approach to the tumor. The resection of these specimens, which are usually discarded, has no functional consequence and, as a common practice, would be performed irrespective of the participation of the patient in the study. Information about tumor characteristics (IDH mutation status, 1p/19q codeletion, WHO grade) was available postoperatively from the histopathological examination performed by an experienced neuropathologist (A.C.), following the WHO 2016 guidelines. The detection of IDH mutations was achieved using immunohistochemistry and high resolution melting (HRM) analysis of isocitrate dehydrogenase 1 (*IDH1*) and isocitrate dehydrogenase 2 (*IDH2*) gene real-time polymerase chain reaction (rtPCR) data [28]. The detection of 1p/19q codeletion was attained using quantitative microsatellite analysis (QuMA) [29]. Patients with a diagnosis different than diffuse glioma were excluded from the study.

### 2.2. Slice Preparation

The specimens were embedded in 2% w/v low melting point agarose and 5 mg/mL glucose in PBS, to retain structural integrity [25], and sliced into constant width samples, appropriate for AFM nanoindentation, with a vibratome (VT1000S, Leica Biosystems, Nussloch, Germany) following a protocol used for similar type experiments [21]. Slicing was performed in constantly carbogenated (95% O_2_, 5% CO_2_) ice-cold (0–4 °C) artificial cerebrospinal fluid (aCSF, containing in mM: 120 NaCl, 2.5 KCl, 26 NaHCO_3_, 2 CaCl_2_, 1 NaH_2_PO_4_, 2 MgCl_2_, 1 Kynurenic acid, and 10 Glucose, pH 7.4) with the following parameters: slice thickness 350 μm, clearance angle 5°, horizontal amplitude 0.8 mm, sectioning frequency 80 Hz, sectioning speed 0.050 mm/s. Slices were transferred on sterile glass coverslips coated with Corning Cell-Tak Cell and Tissue Adhesive (Corning Life Sciences, Amsterdam, The Netherlands).

### 2.3. Elasticity Measurements

The measurements were performed with a Multimode III AFM (Bruker, Billerica, MA, USA) using a liquid cell. SiN cantilevers pre-calibrated by the manufacturer with the Sader method [30] (nominal spring constant *k* = 0.03 N/m, calibrated *k* = 0.05 N/m) having a 25-μm spherical polystyrene bead attached to the tip (Novascan Technologies, Boone, IA, USA) were used. The probe tip sphere diameter was verified by contact mode surface scanning on a tip characterization test grating (TGT1, NT-MDT, Moscow, Russia). Briefly, for each patient, a slice of each tissue type (white matter, glioma) was mounted on the AFM scanner in Leibovitz L-15 medium (Gibco, Thermo Fisher Scientific, Waltham, MA, USA)—a CO_2_ independent medium that can accommodate glioma and white matter tissue for short periods [31]. After allowing the tissue, the medium, and the device to equilibrate for 10 min at 25 °C, the position of the cantilever above the tissue was optically verified. Then, indentations, 100 μm apart from each other, were performed in a 3 × 3 grid at a vertical scan rate of 1.74 Hz, a 2.32 μm ramp size, with an indentation velocity of 8.08 μm/s. A total of 9 force curves per slice were obtained. The last indentation for each slice was repeated on the same position five times at 2-min intervals. All measurements were accomplished within 8 h after tissue acquisition to avoid deterioration of the fresh unfixed tissues. Force curves exhibiting artifacts were discarded.

### 2.4. Force Curve Analysis

Force-distance curves were analyzed with the Nanoscope Analysis 1.5 software provided by the AFM manufacturer (Bruker, Billerica, MA, USA). The raw force curves included a non-contact region and consisted of an approaching and retraction arm. The approaching curves, recorded as deflection *d* of the cantilever versus displacement *z* of the specimen in the vertical axis, were transformed to force *F* = *kd* versus separation *w* = *z* − *d* curves. The elastic modulus was estimated for each approaching force-separation curve through the Hertz model [32] equation
(1)$$F=\frac{4}{3}\frac{E}{1-v^{2}}\sqrt{r}\delta^{3/2}$$
where *v* denotes the Poisson ratio (assumed to be 0.5 [22,23,25,26]), *r* is the tip radius, and *δ* = *w* − *w*_0_ stands for the indentation depth. The contact point location *w*_0_ was treated as a fitting parameter [33], while the lower 20% and the higher 10% of the full force range were ignored in the fitting process to estimate the elastic modulus accurately [21,22].

### 2.5. Statistical Analysis

The conformity of the continuous variables with normal distribution was tested by using the Kolmogorov–Smirnov normality test with the Lilliefors correction. The mean and the standard deviation were obtained for normally distributed variables, while for non-normally distributed variables, the median and the range were reported.

To study the influence of histopathological characteristics on the elastic modulus, the main effects of tissue type (white matter vs. tumor), IDH mutation status (wild-type vs. mutant), and WHO grade (grade II-III vs. IV, grade II vs. III), along with all possible interactions between them were analyzed as fixed effects in a linear mixed-effects model [34] of the elastic modulus (monotonic indentation model), using planned contrasts and grouping measurements by patient. The influence of age was also investigated in the same model as a fixed effect.

To consider the mechanical behavior of tissues under repetitive deformation, the influence of the repetition of the last indentation for each slice on the elastic modulus was analyzed as a fixed effect in a separate linear mixed-effects model (repetitive indentation model), using orthogonal polynomial trends (linear, quadratic) and also grouping by patient. The main effect of tissue type and its interaction with repetition were also investigated as fixed effects.

In both monotonic and repetitive indentation models, sources of inter-patient heterogeneity, including both a random intercept and a random slope for tissue type [35], were studied. Fitting models with a variance structure to account for possible heteroscedasticity across patients was considered. For repetitive indentation only, a first-order autoregressive correlation structure to take into account possible intra-patient dependence of the residual errors was also pursued. The conformity with the assumptions of the linear mixed-effects model theory—i.e., normality and independence of the residuals, as well as of the random coefficients, homoscedasticity, linearity, and no perfect multicollinearity between predictors—was evaluated graphically and, where appropriate, formally. Before analysis, a logarithmic transformation was applied to the elastic modulus, as the latter cannot take negative values.

Reported results were back-transformed to the original scale to facilitate interpretation, and related effects were expressed as ratios of the elastic modulus between contrasting categories. The significance of the variance structures and the correlation structure were tested using the likelihood-ratio test (LRT) with the respective degrees of freedom. The random intercepts and slopes were tested with the LRT against a mixed 0.5 $\chi_{0}^{2}$ + 0.5 $\chi_{1}^{2}$ distribution [36], whereas degrees of freedom for the *F*- and *t*-tests used for the fixed effects were estimated using the Satterthwaite approximation [37,38]. Interaction effects were further investigated with planned contrasts and simple effects analyses of the predictors, while estimated marginal geometric means of the elastic modulus for each predictor level were calculated. Subgroup analyses were planned for tumors of astrocytic and oligodendroglial origin.

The level of statistical significance for all analyses was defined as *p* < 0.05. Statistical analyses were conducted using the R software environment [39] with the packages emmeans [40], nlme [41], nortest [42], and pastecs [43].

## 3. Results

### 3.1. Descriptive Statistics

Twenty patients were initially eligible for participation in the study. One patient refused participation in research and another declined surgery. As a result, both were excluded from enrollment. Of the eighteen patients enrolled, one was excluded from further analysis because the tumor specimen was insufficient for the study and another because a diagnosis of a WHO grade I glioma was made.

Data from the remaining sixteen patients, ten male and six female, with a median age of 50.5 (range 17–77) years, were analyzed. An *IDH1/2* mutation was detected in tumors from four patients, while twelve patients had an IDH-wildtype tumor. Five patients harbored a WHO grade II diffuse astrocytoma, two an anaplastic astrocytoma, and nine a glioblastoma. No oligodendroglial tumor was detected. The tumor resided in the left hemisphere in four patients and the right one in twelve (Table 1). In total, 293 measurements at different points across the surface of the slices were performed. The median value of the elastic modulus was 209 (range 64–1170) Pa for peritumoral white matter and 175.5 (range 51–1110) Pa for tumor tissue. Another 129 measurements were performed to study the influence of repetitive indentation.

### 3.2. Relation with Patient and Tumor Characteristics—Monotonic Indentation Model

In the monotonic indentation model, the relationship between *E* and tissue type showed considerable inter-patient heterogeneity, as intercepts and slopes (for both *p* < 0.001) varied significantly across patients. A variance structure to account for heteroscedasticity among patients was employed (*p* < 0.001).

The main effects of tissue type, age, IDH mutation status, and WHO grade, as well as the tissue type by IDH mutation status interaction effect, were non-significant (Table 2).

In contrast, the IDH mutation status by WHO grade interaction effect was significant, indicating that the effect of IDH mutation status on tissue elasticity was different between patients with a WHO grade II diffuse astrocytoma and a WHO grade III anaplastic astrocytoma. Therefore, the tissue elasticity was similar between patients with an IDH-mutant and an IDH-wildtype diffuse astrocytoma (ratio = 0.82, 95% CI 0.45–1.49, *p* = 0.480). This pattern was independent of tissue type (white matter: ratio = 0.61, 95% CI 0.27–1.37, *p* = 0.221; tumor: ratio = 1.09, 95% CI 0.49–2.47, *p* = 0.823). Tissues from IDH-mutant anaplastic astrocytoma cases were significantly stiffer than those from IDH-wildtype ones, when tissue type was ignored (ratio = 2.29, 95% CI 1.00–5.22, *p* = 0.0496) (Figure 1). A similar, although non-significant, proportional difference in elasticity was observed between IDH-mutant and IDH-wildtype anaplastic astrocytoma cases in both white matter (ratio = 2.27, 95% CI 0.69–7.50, *p* = 0.173) and tumor tissue (ratio = 2.30, 95% CI 0.70–7.60, *p* = 0.164).

The effect of tissue type was similar between IDH-wildtype glioblastoma patients, where the elasticity was comparable between tumor and peritumoral white matter (ratio = 0.89, 95% CI 0.57–1.39, *p* = 0.593), and the average of IDH-wildtype WHO grade II and III astrocytoma cases. However, the effect of tissue-type differed between diffuse astrocytoma and anaplastic astrocytoma patients, although not significantly. Looking at each simple effect, in WHO grade II cases, the elasticity of tumor was similar to that of peritumoral white matter (ratio = 1.24, 95% CI 0.68–2.25, *p* = 0.462), but in WHO grade III cases the tumor was non-significantly softer than white matter (ratio = 0.43, 95% CI 0.17–1.08, *p* = 0.070) (Figure 2).

The initially planned subgroup analyses were not performed, as no oligodendroglial tumor was detected.

### 3.3. Repetitive Indentation Model

In the repetitive indentation model, the relationship between the elastic modulus and tissue type showed considerable inter-patient heterogeneity, as well as intercepts and slopes (both *p* < 0.001) varying significantly across patients. Both a variance structure to correct for heteroscedasticity among patients (*p* < 0.001) and a first-order autoregressive correlation structure (*p* = 0.006) were employed.

There was a significant main effect of repetitive indentation, as both linear and quadratic trends of repetition were significant. However, the main effect of tissue type was not significant (Table 3). Most notably, the linear trend was steeper in tumors than in brain, even though the quadratic trends were similar. Indeed, looking at simple effects, in peritumoral white matter tissue the linear trend was non-significant (ratio = 1.00, 95% CI 0.91–1.10, *p* = 0.942), in contrast to the quadratic one which was significant (ratio = 0.88, 95% CI 0.81–0.95, *p* = 0.002). However, in tumor tissue both the linear (ratio = 1.17, 95% CI 1.06–1.28, *p* = 0.003) and the quadratic trend of repetition (ratio = 0.90, 95% CI 0.83–0.98, *p* = 0.015) were significant (Figure 3).

## 4. Discussion

In the present study, tissues from IDH-mutant WHO grade III astrocytomas were found to be 129% stiffer than those from IDH-wildtype cases. This finding was also observed with a similar proportional difference when peritumoral white matter and tumor tissue were considered separately, although the loss of power prevented it from showing statistical significance. An MRE study also found IDH1-R132C-mutant WHO grade III gliomas to be stiffer than IDH1-wildtype ones [16]. On the contrary, a previous AFM indentation study found IDH1-R132C-mutant WHO grade II and III gliomas to be softer than IDH1-wildtype ones [27]. However, the latter study was performed on fresh frozen tissues and it also showed that the elastic modulus of fresh tissues differed from that of frozen ones. Instead, the present study measured tissue elasticity on fresh tissues to establish a more accurate approximation of the in vivo elasticity, with the drawback, however, of time constraints.

Tissue elasticity was similar between IDH-mutant and IDH-wildtype diffuse astrocytomas, while the sample of Pepin et al. [16] did not recruit any patient with an IDH-wildtype diffuse astrocytoma and, therefore, was unable to study the effect of IDH mutation status in WHO grade II tumors. However, they were able to study this effect in glioblastomas, and they observed that IDH1-mutant tumors were on average stiffer than IDH1-wildtype ones. In contrast, Miroshnikova et al. [27] found IDH1-mutant glioblastomas to be softer than IDH1-wildtype ones, with an elasticity comparable to that of IDH1-mutant WHO grade II and III tumors. Neither of these studies sought rare *IDH1/IDH2* mutations, thus adding to the uncertainty of their findings. The present study was not able to enroll any patient with an IDH-mutant glioblastoma, although it seems that the investigation of the mechanical properties of these rare tumors could provide further evidence about tumor progression.

Despite the fact that IDH mutations appear early in gliomagenesis [44], it remains unclear whether they initiate a less aggressive pathway to tumor development in comparison to IDH-wildtype gliomas [45] or they represent a tumor-suppressive factor that counterbalances glioma tumorigenesis and progression [46]. The IDH mutations present in diffuse gliomas are of the gain-of-function type and result in the production of 2-hydroxyglutarate (2-HG) [47], a metabolite inhibiting enzymes that prevent histone and DNA hypermethylation [48], leading to extensive DNA and histone methylation [49,50], associated with favorable patient outcomes [51]. Furthermore, although initial research linked 2-HG with the upregulation of hypoxia-induced factor 1α (HIF-1α) [52,53], a transcription factor activated by hypoxia and associated with tumor neovascularization, viability and growth [54], later studies questioned this link supporting the downregulation of HIF-1α by 2-HG [55,56] or supported a conditional regulation of HIF-1α depending on tissue stiffness [27]. It should also be noted that interstitial hypoxia, a key feature of tumor progression, is correlated with glioma grade and aggressiveness [57]. Many aspects of the interplay between hypoxia, IDH mutation status, glioma progression, and tissue elasticity remain obscure [27,45]. In view of these considerations, the exact mechanisms that affect the mechanical properties of the microenvironment and result in the differences in elasticity observed between WHO grade III cases but not between WHO grade II cases, as well as their implications on tumor aggressiveness, are still not fully clarified [16]. Nevertheless, IDH-mutant WHO grade III astrocytomas show a much more favorable profile than their IDH-wildtype counterparts in terms of prognosis [58] and feasibility of gross total resection [59]. The possibility to determine IDH mutation status preoperatively using biomechanical methods, like MRE [16], would assist surgical and medical treatment planning for such cases.

Moreover, in the present work, in WHO grade II cases, the elasticity of tumor tissue was found similar to that of peritumoral white matter. Results from elastography seem to be contradictory, as some studies found tumor tissue to be stiffer than white matter in low-grade cases [13,14], and others found it softer across all grades [16]. However, this inconsistency seems to be removed when considering that the latter study compared the tumor with the contralateral hemisphere white matter. It has been proposed that peritumoral white matter presents differences in elasticity from normal white matter, due to edema and compression by the lesion [15]. The results of a recent semi-quantitative ultrasound elastographic study agree with this hypothesis, as the tumor core was found slightly stiffer than the tumor periphery, which was slightly stiffer than the peritumoral white matter, and the latter was substantially softer than distant white matter [12]. Moreover, diffuse glioma cells have the propensity to invade adjacent brain tissue and to migrate along white matter tracts and perivascular spaces [60]. It should be noted that in the present study, tumor cell infiltration has not been quantified in the white matter specimens. Nevertheless, exploring the possible association of peritumoral white matter elasticity with prominent traits of its histology, like tumor cell infiltration, as well as myelin and hyaluronan [61] content and properties, factors that play a key role in the interaction between glioma and extracellular matrix [62,63,64], would be an interesting topic for future research.

Most elastographic studies combined findings from WHO grade III and IV gliomas [12,13,14] and found that in ‘high-grade gliomas’, the whole tumor tissue was either softer [14] or non-significantly stiffer [13] than peritumoral white matter. Moreover, Cepeda et al. [12], also combining WHO grade III and IV tumors, found that the tumor core was substantially softer than the tumor periphery, which, in turn, was slightly softer than the peritumoral white matter. In the present study, the tumor was found softer than the peritumoral white matter in WHO grade III cases, although non-significantly. Anaplastic astrocytomas are characterized by hypercellularity [65] and ECM remodeling [66]. Although glioma cells have been found to be stiffer than normal astrocytes [67,68], it has been observed that cancer cells are, in general, softer than their respective ECM [69,70]. Although this has not yet been proved in gliomas, if that is the case, a radical increase in cell proliferation has the potential to decrease tissue stiffness in comparison to normal tissue. Furthermore, glioma cells interact with the ECM [71,72], as they produce proteases that decompose ECM constituents to enhance their migration [73], while they deposit a modified ECM that serves as a substrate [74]. The result of these processes is the structural degradation and disruption of tissue mechanical homeostasis [72], reflected in the (non-significant) softening of anaplastic astrocytoma tissue on average as compared to peritumoral white matter.

In WHO grade IV cases, tumor elasticity was found similar to that of peritumoral white matter in the present study. Glioblastoma periphery histopathology is characterized by hypercellularity [75], intratumoral thromboses [76] and necrotic regions [75], while ECM remodelling is even more evident [77]. Blood clots [78] are generally stiffer than average glioma tissue, while necrotic tissue has been softer than non-necrotic ones [26]. Glioma vascularity may also affect tissue elasticity, as blood vessels are stiffer [79] than either average glioma tissue or white matter. It is established that diffuse gliomas, like most solid tumors [80], show an angiogenic behavior [81], which is more evident in glioblastomas [82]. In the present study, tumor or white matter vascularity has not been systematically assessed in a quantitative manner. However, tumor vascularity was qualitatively assessed during the histopathological examination of the tumors and it was found consistent with tumor grading [65]. White matter histopathological examination did not make any conspicuous observation regarding tissue vascularity. The result of the complexity of the factors at play in the case of glioblastomas is considerable intra-tumor mechanical heterogeneity [19] and equivocal results between studies [12,13,14] or even within the same study [19]. In MRE studies (which average over the entire tumor volume) increased glioma WHO grade is consistently associated with decreased tumor elastic modulus [16,17,18]. More insight could be gained into the relationship between WHO grade and elasticity by further studies addressing the individual contribution of each histopathological feature.

Furthermore, it has been observed that white matter stiffness is positively correlated with myelin content [83]. Nevertheless, although acute demyelination has been associated with decreased stiffness, chronic acquired demyelination has been connected to increased stiffness [84], while inherited hypomyelination does not affect tissue stiffness [85]. In the present study, the white matter specimens were only examined to verify that they truly represent white matter tissue and their content in myelin was not quantitatively determined. However, none of the patients was known to suffer from a demyelinating disease. Furthermore, although advanced age is related to white matter demyelination [86,87], age was not associated with tissue elasticity in the sample of the present work. These results agree with observations on mammalian brain tissue, where white matter elasticity did not differ between young and old subjects, with the exception of striatal white matter [85].

In the repetitive indentation model, although both tumor and peritumoral white matter showed initial stiffening followed by softening with repetitive indentations, it seems that for the number of repetitions studied in this work, stiffening was fully reversed in white matter, while only partially reversed in tumors. Although the effects were not large, when considering measurement accuracy, even small deviations from the true value of the measured quantity are important. It follows that further research on glioma and/or white matter elasticity using nanoindentation should take into account the effects of preconditioning. Therefore, the difference in the mechanical behavior between white matter and tumor tissue is worth investigating further.

It should also be pointed out that the heterogeneity between subjects in both monotonic and repetitive indentation models was considerable. This heterogeneity could be attributed to various factors not studied yet, for which future research on glioma elasticity should consider and further identify. In this context, a few remarks on mathematical modeling are in order. Apart from non-linear elasticity theory (which was long ago known in the literature), reference is made to gradient elasticity theory [88,89,90] for linear heterogeneous elastic materials. Within a one-dimensional version, this theory reads *σ* = *E*[*ε* − *l^2^*∂^2^*ε*/∂^2^*x*], where the second term on the right-hand side is an enhancement of linear elasticity, with *l* being an internal material length related to the underlying heterogeneity. Employment of this gradient elasticity relation can lead to a modification of Equation (1). which, in turn, can be used for interpreting *F*-*δ* curves, which will now depend on the newly introduced heterogeneity parameter *l*. Thus, measurements of the elastic modulus *E*, would now depend on the particular microstructural heterogeneity of the specimen tested through the internal length *l*. Such type of considerations will be a subject of future studies.

Obviously, the present work meets certain limitations. The sample size of the study is certainly small, even if it is comparable to that of studies that investigated similar effects [16,19,26,27]. Apart from the relatively low statistical power of the present study to detect true effects, no IDH-mutant glioblastomas were studied. As a result, the effect of IDH mutation status on tissue elasticity could not be investigated in glioblastomas. Moreover, very few patients with an anaplastic glioma participated, a fact that may compromise the ability of the study to provide definite conclusions on WHO grade III astrocytic tumors. Nevertheless, it should be noted that these are quite rare tumors, and a prospective observational study cannot detect a priori factors that become known only later in its course. Additionally, the present work should ideally be viewed as a first effort to unravel a puzzle that will be completed with further similar medium-to-large sample size replication studies from multiple centers to minimize sources of possible bias and provide more reliable results. Further studies with a larger sample would achieve higher power and could also examine the possible effect of 1p/19q codeletion and other factors that may influence the mechanical behavior of diffuse gliomas.

The present study recruited patients between 2017 and 2019, therefore, tumor identification followed the 2016 WHO guidelines for the classification of tumors of the central nervous system, which did not designate any diffuse pediatric-type glioma diagnostic category apart from the diffuse midline glioma, H3 K27M-mutant [65]. All necessary molecular characterization for a hemispheric non-midline diffuse glioma following the 2016 guidelines was performed, and immunohistochemical and molecular analysis was planned for the necessary genes/chromosomes. However, a systematic search for a wide array of gene mutations, such as histone mutations in non-midline tumors, was not yet justified. Four patients in this study were <30 years old, all harboring IDH-wildtype tumors (patients 1, 4, 13, and 16, Table 1). The possibility that these tumors belonged to a newly defined diagnostic category of diffuse pediatric-type low-grade or high-grade glioma [91] cannot be excluded, a hypothesis requiring further immunohistochemical and molecular analyses to be verified or rejected. Future studies, following the 2021 WHO guidelines, should address this issue and classify cases using a wider molecular profiling strategy.

The elasticity measurements conducted ex vivo on fresh unfixed tissue slices should be considered as a realistic approximation of direct in vivo tissue elasticity. Ideally, the measurements should be done in situ, which does not lie within the capabilities of the present technology. In addition, only nine measurements per slice were performed due to time limits, as the tissues were not fixed, and the present study did not aim to map the elasticity across the surface of each slice but to determine its association with histopathological features. Currently, AFM setups with a *z* scan range that is > 100 μm, which improves measurement performance, are still not universally available. Further optimization, standardization, and automatization of all methods from harvesting to measurement would enable the attainment of a shorter total examination time per specimen and, consequently, higher statistical power, better specimen quality, and more measurements per specimen. A higher number of measurements would achieve better precision, even though a large intra-patient heterogeneity has been found by studies with more measurements per tissue as well [19,26]. Likewise, time limits prevented elasticity measurements on multiple slices per patient for each tissue type. Future advances in technology could also achieve multiple serial slice measurements and enable the study of elasticity in three dimensions.

Owing to its versatility, AFM nanoindentation is not yet a standardized procedure. However, efforts to minimize confounding effects—due to substrate, low/high indentation speed, tip shape, osmolarity, and pH—and achieve results as close as possible to the true elastic modulus have already been made. The Hertz model, used to estimate the elastic modulus from the recorded force curves, assumes linear elasticity, even though it is well known that tissues are non-linear elastic and even more complex materials. Nevertheless, linear elasticity is a widely used approximation at low stress levels. All experimental parameters, like tip radius, indentation depth, and distance between indentation sites, were chosen to fulfill model assumptions. It should be emphasized that the values obtained for the elastic modulus of white matter tissue were comparable to those observed in mammals using the same method [21,24].

To the best of our knowledge, this study is the first to investigate the interaction effect between glioma IDH mutation status and WHO grade on the elastic modulus for both tumor tissue proper and peritumoral white matter using AFM nanoindentation. Moreover, this study appears to be the first to systematically determine the elastic modulus of human white matter using AFM nanoindentation with spherical tips on fresh tissue slices ex vivo.

## 5. Conclusions

In summary, tissues from IDH-mutant cases were found significantly stiffer than those from IDH-wildtype ones among anaplastic astrocytoma patients but similar in elasticity to IDH-wildtype cases among diffuse astrocytoma patients. The tumor periphery was found similar in elasticity to that of adjacent white matter in WHO grade II and IV tumors, while non-significantly softer in WHO grade III tumors. During repetitive indentation, both tumor and peritumoral white matter showed initial stiffening followed by softening, but stiffening was fully reversed in white matter, while only partially in a tumor.

These results present preliminary but definite evidence that the mechanical properties comprise a phenotypic characteristic related to histopathological traits of diffuse gliomas. Intrinsic brain tumor mechanobiology has the potential to offer insight into the mechanisms of gliomagenesis, a novel view on glioma biology, and enhanced tools for the clinical practice of neuro-oncology. To take full advantage of the mechanical phenotype of gliomas, sources of inherent heterogeneity across patients and within each patient should be sought by further research implementing larger samples and investigating more factors possibly influencing glioma tissue mechanics.

## Figures and Tables

**Figure 1 cancers-13-04539-f001:**
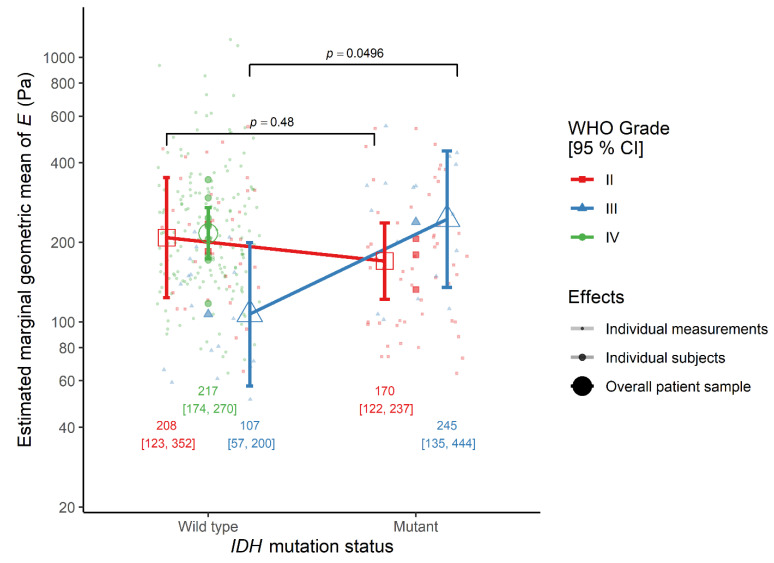
The interaction effect between IDH mutation status and WHO grade on the elastic modulus *E* in the monotonic indentation model. IDH, isocitrate dehydrogenase gene family; WHO, World Health Organization.

**Figure 2 cancers-13-04539-f002:**
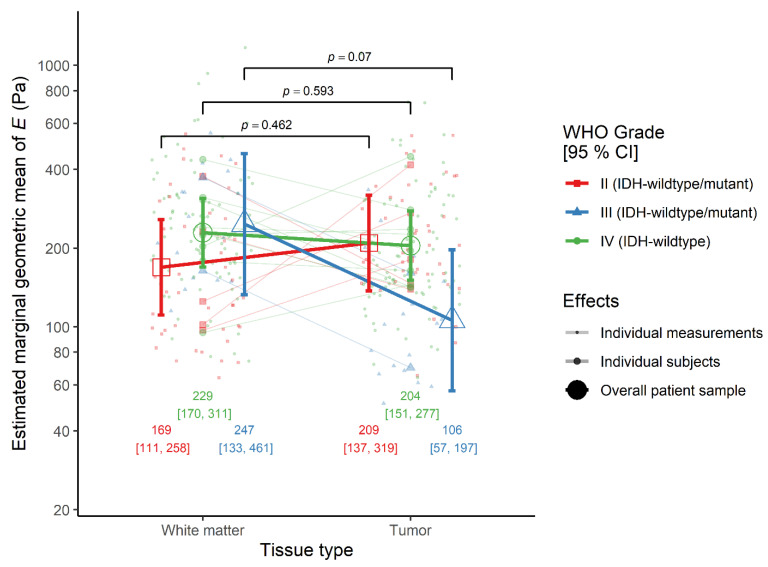
The interaction effect between tissue type and WHO grade on the elastic modulus *E* in the monotonic indentation model. IDH-wildtype.

**Figure 3 cancers-13-04539-f003:**
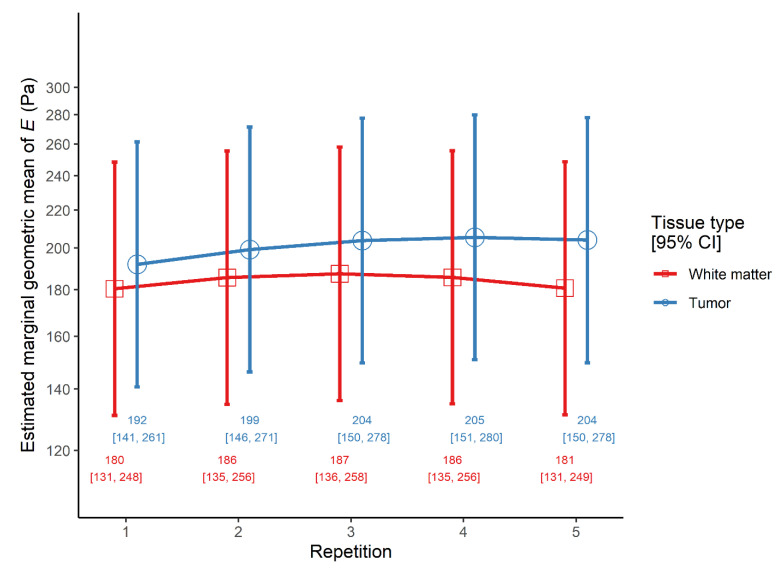
The interaction effect between tissue type and repetition on the elastic modulus *E* in the repetitive indentation model.

**Table 1 cancers-13-04539-t001:** Patient and tumor characteristics.

No.	Gender	Age (Years)	Side	Lobe(s)	IDH Mutation Status	Grade
1	Female	27	Right	Parietal	Wild-type	III
2	Male	77	Right	Parietal, Occipital	Wild-type	IV
3	Male	36	Left	Frontal, Temporal, Insula	Mutant	II
4	Female	69	Left	Temporal	Wild-type	IV
5	Male	17	Right	Frontal	Wild-type	II
6	Male	36	Right	Frontal, Parietal	Mutant	II
7	Female	46	Right	Frontal	Wild-type	IV
8	Male	75	Right	Temporal	Wild-type	IV
9	Female	64	Left	Temporal	Wild-type	IV
10	Male	68	Right	Temporal, Parietal	Mutant	II
11	Female	55	Right	Frontal, Parietal	Wild-type	IV
12	Male	61	Left	Temporal, Parietal	Wild-type	IV
13	Female	25	Right	Temporal	Wild-type	IV
14	Male	55	Right	Temporal, Occipital	Wild-type	IV
15	Male	39	Right	Frontal	Mutant	III
16	Male	24	Right	Frontal, Temporal, Insula	Wild-type	II

IDH, isocitrate dehydrogenase gene family.

**Table 2 cancers-13-04539-t002:** Fixed effects in the linear mixed-effects model of the association between the elastic modulus and tumor characteristics (monotonic indentation model).

Effect	Contrast	Ratio	95% CI	*p*
Tissue type	White matter vs. Tumor tissue	0.76	0.48, 1.19	0.211
Age (years)		1.00	0.99, 1.01	0.980
IDH mutation status	Wild-type vs. mutant IDH	1.57	0.57, 4.31	0.359
WHO grade				0.532
	Grade II-III vs. IV	1.41	0.72, 2.79	0.300
	Grade II vs. III	0.74	0.28, 1.95	0.518
IDH mutation status × WHO grade	Wild-type vs. mutant IDH × Grade II vs. III	2.80	1.05, 7.48	0.041
Tissue type × IDH mutation status	White matter vs. tumor tissue × Wild-type vs. mutant IDH	1.56	0.16, 14.85	0.684
Tissue type × WHO grade				0.157
	White matter vs. tumor tissue × Grade II-III vs. IV	1.38	0.45, 4.29	0.555
	White matter vs. tumor tissue × Grade II vs. III	0.12	0.01, 1.09	0.058
Tissue type × IDH mutation status × WHO grade	White matter vs. tumor tissue × Wild-type vs. mutant IDH × Grade II vs. III	0.56	0.06, 5.11	0.590

For each main effect, the proportional change in elastic modulus per year (for the effect of age) or between contrasting categories of the factor (for tissue type, IDH mutation status, and WHO grade) is reported. For each interaction term, the ratio compares the effects of a factor between contrasting categories of another factor. IDH, isocitrate dehydrogenase gene family; WHO, World Health Organization.

**Table 3 cancers-13-04539-t003:** Fixed effects in the linear mixed-effects model of the association between the elastic modulus and repetitive indentation (repetitive indentation model).

Effect	Contrast	Ratio	95% CI	*p*
Tissue type	White matter vs. tumor tissue	1.09	0.65, 1.83	0.723
Repetition				<0.001
	Linear trend	1.13	1.02, 1.26	0.021
	Quadratic trend	0.85	0.79, 0.92	<0.001
Tissue type × Repetition				0.065
	White matter vs. tumor tissue × Linear trend	1.27	1.03, 1.57	0.026
	White matter vs. tumor tissue × Quadratic trend	1.04	0.89, 1.21	0.646

For tissue type, the proportional change in elastic modulus between categories of the factor is reported. Ratios for trends are indicative of the overall proportional change in the elastic modulus across repetitions (linear trend) and the proportional change of the instantaneous proportional change of the elastic modulus per repetition (quadratic trend), respectively. For each interaction term, the ratio compares the effects of each trend between categories of tissue type.

## Data Availability

The data presented in this study are available on request from the corresponding author. The data are not publicly available due to privacy restrictions.

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
