# Peer review of "Atomic Force Microscope Nanoindentation Analysis of Diffuse Astrocytic Tumor Elasticity: Relation with Tumor Histopathology"

_cancers, 2021, doi:10.3390/cancers13184539_

Round 1
Reviewer 1 Report
In their manuscript, “Atomic Force Microscope Nanoindentation Analysis of Diffuse Astrocytic Tumor Elasticity: Relation with Tumor Histopathology,” Tsitlakidis et al. investigate the fresh tissue elasticity of infiltrating astrocytomas in tumor and adjacent brain parenchyma.
A few questions arise:
For those patients who are young (20s) with wildtype IDH, please provide further molecular characterization, if possible, or a caveat. Could these tumors belong to the new diagnostic category, Diffuse pediatric-type high-grade glioma, H3-wildtype, and IDH-wildtype?
Aside from the statistical significance, are the elasticity differences between the grade 3 cases biologically significant?
How much tumor infiltration was there in the different white matter specimens? How does amount of infiltration affect elasticity? This should also be incorporated into the discussion, where it is noted that peritumoral white matter may be different from normal white matter due to edema and compression (p. 10 line 343).
Can histological correlation be made with these measurements to remove the speculation from the discussion regarding the histological features and their contribution to tissue softening (p. 10 lines 356-362)? For each grade, different histologic features may be present, so being more specific may yield additional information.
Minor: Please read for syntax and standardization of italicization of, for example, “IDH.”
Reviewer 2 Report
Comments:
It is an interesting article proposing to exploit the mechanical property difference between the tissues from high- and low-grade brain tumors (WHO classified I-IV) and between the tumor and peritumoral white matter region tissues and based on IDH mutant/wild type sub types. The peritumoral white matter region was shown to be tighter with a higher elastic modulus. In particular, in grade III tumor, the tumor tissue is non-significantly softer than white matter. Probably increasing the number of experimental subjects and tissues, statistical significance could be achieved. The uniqueness of the study depends on investigating the elastic modulus of the tumor tissue and adjacent white matter in combination with IDH wild type/mutant. The authors were able to demonstrate significance in Grade III tumor, for the IDH mutant type to be tighter than IDH wild type (Figure 1)
- The myelination of nervous system in the white matter region is known to cause tissue stiffness. Was any attempt made by the authors to investigate the brain sections by histology or EM to compare their contribution to the observed values (elastic modulus)?
- Similarly, tumors are in general vascular. Was any attempt made to investigate the brain sections by histology or EM? Or any attempt made to discuss based on existing literature?
- In the repetitive indentation model, elastic modulus values were shown to be distinctly different between linear trend and quadratic trend model. Can the authors explain how they arrived at the values?
Round 2
Reviewer 1 Report
The authors have addressed all of the points with discussion and caveats, but the weaknesses remain in the paper. In particular, the significance of these results remains obscure, with only a suggestion that MRE could be used preoperatively to determine IDH mutational status. It seems doubtful that the slight difference in elasticity between tumor types, which is only significant for grade III astrocytomas, would be detectable by MRE, and this would require establishment of grade first. These concerns are amplified by the lack of understanding of the histology (tumor microenvironment, infiltrative burden of tumor in the white matter, etc.). In addition, the machine learning field is already making progress on this problem, and algorithms have been developed on MRI studies that have some value in determining IDH mutational status.
However, given the recalcitrance of gliomas to treatment, this type of data regarding a physical property of the tumors may be useful as a starting point for further studies, understanding, and treatment.